# The prevalence of age-related macular degeneration and osteoporosis in the older Polish population: Is there a link?

Agnieszka Budnik[1], Marcin Palewski[1], Magdalena Michnowska-Kobylińska[1], Łukasz Lisowski[1], Magda Łapińska[2], Zofia Stachurska[3], Anna Szpakowicz[4], Jerzy Konstantynowicz[5], Karol Kamiński[2], Joanna Konopińska [1] *

1 Department of Ophthalmology, Medical University of Białystok, Białystok, Poland, 2 Department of Population Medicine and Lifestyle Diseases Prevention, Medical University of Białystok, Białystok, Poland, 3 Population Research Centre, Medical University of Białystok, Białystok, Poland, 4 Department of Cardiology, Medical University of Białystok, Białystok, Poland, 5 Department of Pediatrics, Rheumatology, Immunology and Metabolic Bone Diseases, Medical University of Białystok, Białystok, Poland

* joannakonopinska@o2.pl

**Data Availability Statement:** All relevant data are available on Harvard Dataverse: https://dataverse.

## Abstract

### Background

Age-related macular degeneration is the primary cause of irreversible blindness in developed countries, whereas the global prevalence of osteoporosis–a major public health problem–is 19.7%. Both diseases may coincide in populations aged >50 years, leading to serious health deterioration and decreased quality of life.

### Objectives

This study aimed to analyze the relationship between age-related macular degeneration and osteopenia, defined as decreased bone mineral density, in the Polish population.

### Methods

Participants were derived from the population-based Bialystok PLUS Study. Randomized individuals were stratified into two groups, those with age-related macular degeneration (AMD-1 group) or without age-related macular degeneration (AMD-0 group). Using a cutoff value of −1.0 to identify low bone mass, participants with femoral bone mineral density T-scores above −1.0 were assigned to the normal reference, and those with T-scores below −1.0 were assigned to the osteopenia category. Among 436 Caucasian participants aged 50–80 years (252 women, 184 men), the prevalence of age-related macular degeneration was 9.9% in women and 12.0% in men. Decreased bone mineral density based on T-scores was observed in 36.9% of women and in 18.9% of men. Significant differences in femoral bone mineral density between the AMD-0 and AMD-1 groups were detected only in men (mean difference [95% confidence interval] = 0.11 (0.02; 0.13); p = 0.012 for femoral bone mineral density, and 0.73 [0.015; 0.94]; p = 0.011 for the femoral T-score). No associations were observed between bone mineral density and age-related macular degeneration in women.

harvard.edu/dataset.xhtml?persistentId=doi:10.7910/DVN/TZVNHA.

**Funding:** This study was supported by funds of the Ministry of Science and Higher education (Poland) within the project "The Excellence Initiative - Research University" (proteomic research), and from statutory funds of Medical University of Bialystok.

**Competing interests:** The authors have declared that no competing interests exist.

## Conclusion

Decreased femoral bone mineral density may be associated with a higher risk of age-related macular degeneration in men, but a causal link remains unclear.

## Introduction

Age-related macular degeneration (AMD) is the primary cause of irreversible blindness in developed countries [1]. The number of European patients with AMD is projected to increase as the population ages [2].

AMD primarily affects the macula, leading to vision loss in the central 15 to 20 degrees of the visual field, often in both eyes. Not surprisingly, patients with AMD frequently sustain injuries, including skeletal trauma and fragility fractures [3, 4]. These fractures may be at least partly associated with (or caused by) age-related bone loss and osteoporosis. An association between hip bone mineral density (BMD) and the risk of AMD was observed in female participants in the Study of Osteoporotic Fractures [5]. However, concomitant osteoporosis was not associated with other age-related ocular disorders, such as cataracts, open-angle glaucoma, or diabetic retinopathy in a Korean study [6].

In 2010, the prevalence of osteoporosis among 50- to 84-year-old women from the European Union was estimated at 21%, compared with only 6% among men [7]. However, a growing body of evidence suggests that the prevalence of male osteoporosis may be underestimated [8]. Clinically, osteoporosis manifests as a higher risk of skeletal fragility and frequently leads to premature mortality. Every third woman and every fifth man over the age of 50 years in developed countries are estimated to experience a bone fracture during their remaining lifetime [3, 9, 10]. Furthermore, a two-fold higher risk of falls was observed in older women with AMD than in those without AMD in a prospective cohort study [11].

We could identify only two published studies revealing an epidemiological link between osteoporosis and AMD [4, 6]. Given their importance to public health, assessments of the coincidence of osteoporosis and AMD and underlying mechanisms thereof are warranted. The objective of this study was to analyze the relationship between osteoporosis and AMD.

## Materials and methods

### Study design and participants

The study included participants of the Bialystok PLUS Study, who were 50–80 years of age. The Bialystok PLUS Study was a prospective, population-based, cohort study in which the determinants and occurrence of cardiovascular, neurological, ophthalmic, psychiatric, musculoskeletal, and endocrine diseases were analyzed among residents of the city of Bialystok (Northeastern Poland). It had a fully randomized design. The aims, methodology, and protocol of the study have been described in detail elsewhere [12]. The study data were collected from November 2018 to June 2021. The minimum age limit was set at 50 years, in line with the diagnostic criteria for AMD.

### Bone mineral density measurement

The analyzed parameters included the areal BMD of the total hip, femoral neck, lumbar spine, and total body/whole skeleton. The BMD was measured using dual-energy X-ray absorptiometry (Lunar iDXA; GE Healthcare, Madison, WI, USA). Standard densitometric procedures

and quality assurance were performed, and standard scanning conditions were used, according to official recommendations. The percent coefficient of variation of the dual-energy X-ray absorptiometry method was 2.8% for the total femur measurements. Individuals with femur T-scores of −1.0 and above were assigned to the normal T-score group, and those with T-scores below −1.0 to the low T-score group. An operational diagnosis of osteoporosis was made based on the International Society for Clinical Densitometry standard guidelines. However, the cutoff T-score of −1.0 was intentionally used for the stratification to prevent underestimation, and to identify participants at increased risk of low bone mass.

## Diagnosis of AMD

Fundus photography was performed on all study participants without previous pharmacological mydriasis by using a 35° color digital fundus camera (Canon CR-2 PLUS AF; Canon U.S. A., Inc., New York, USA). Fundus images were graded according to the Wisconsin AMD grading system [13] and the modified International Classification System [14] by appropriately trained retinal specialists (ŁL, MMK). The eyes of each participant were separately graded and classified and the eye with the more severe grade was selected for further analyses. Similar to a previous population-based study [15], we distinguished between early and late AMD. We also analyzed the presence of AMD-specific lesions, such as retinal pigmentary alterations, large drusen ($\geq$125 μm), and a large drusen area ($\geq$331 820 μm$^2$) [16], as a separate outcome variable. Patients were assigned to one of two groups based on the presence (AMD-1 group) or absence (AMD-0 group) of AMD.

## Other characteristics and methods

Anthropometric measurements were performed in a standard manner using certified equipment. Body weight and height were measured with the participants barefoot and wearing lightweight indoor clothing. Body mass index was calculated as weight (kg)/(height [m])$^2$. Fasting blood samples were collected from all participants to determine their serum concentration of 25-hydroxyvitamin D (25[OH]D) by using a gamma counter (1470 Wizard; Perkin-Elmer, Turku, Finland) and radioimmunoassay (DiaSorin, Stillwater, MN, USA).

## Statistical analysis

Statistical analyses were performed using the R statistical package (version 4.1.3), with the α-value set to 0.05. Quantitative variables were compared between groups using the Mann–Whitney U-test or Student t-test for independent samples, as applicable. The normal distribution of the study variables was verified using the Shapiro–Wilk test, and the homogeneity of their variances was verified using Levene's test. Mean differences (MDs) and odds ratios (ORs) were calculated, along with their 95% confidence intervals (CIs). The relationships between categorical variables were analyzed using the chi-square test. Univariate and multivariable logistic regression models were created using AMD as the outcome variable. The results were presented as ORs with 95% CIs. To verify the significance and goodness of fit of the models, the chi-square statistic and Nagelkerke pseudo-$R^2$ were calculated. Nagelkerke pseudo-$R^2$ may take values from 0 to 1, the higher the value, the better quality of the model. The significance of the relationship was verified with the chi-square test (OR 95% CI, odds ratio with a 95% confidence interval).

The study was conducted in accordance with the provisions of the Declaration of Helsinki, and all participants provided written informed consent prior to enrollment. The Bioethics Committee of the Medical University of Białystok approved the study protocol (approval number R-I-002/108/2016 of 31 March 2016).

                                                  

## Results

The study sample consisted of 436 patients aged 50–80 years (252 women). Detailed characteristics of the study participants are listed in Table 1. AMD was diagnosed in 47 patients (10.8%). The prevalence of AMD in women and men were 9.9% and 12.0%, respectively. Decreased BMD based on T-scores was observed in 36.9% of women and 18.9% of men. The group of women included seven patients (five in the AMD-0 group and two in the AMD-1 group) who received treatment for osteoporosis in the form of bisphosphonates (n = 6), calcium supplements (n = 3), and/or vitamin D (n = 5).

Women in the AMD-0 group were younger than those in the AMD-1 group, and the two groups did not differ in terms of body mass index, height, body weight, or serum 25(OH)D concentration. Men were homogenous in terms of age and biometric parameters.

Significant differences in femoral BMD and femur T-scores between the AMD-0 and AMD-1 groups were observed only in men (MD = 0.11 [0.02; 0.13]; p = 0.012 for femoral BMD and MD = 0.73 [0.015; 0.94]; p = 0.011 for femur T-score). The parameters of femoral neck BMD were worse in the AMD-1 group than in the AMD-0 group, with the between-group difference nearing the statistical significance. No association was observed between BMD and AMD in women. 25(OH)D status was not associated with the prevalence of AMD or the risk of a low femoral BMD in men or women.

**Table 1. Demographic, clinical, and densitometric characteristics of study participants.**

|  | Overall (n = 436) | | | Women (n = 252) | | | Men (n = 184) | | |
|---|---|---|---|---|---|---|---|---|---|
|  | AMD-0 (n = 389) | AMD-1 (n = 47) | p | AMD-0 (n = 227) | AMD-1 (n = 25) | p | AMD-0 (n = 162) | AMD-1 (n = 22) | p |
| Age, years | 63.00 (58.00; 68.00) | 66.00 (59.00; 69.00) | 0.100 | 62.00 (58.00; 67.50) | 67.00 (63.00; 72.00) | **0.008** | 63.00 (56.00; 69.00) | 64.50 (56.25; 66.00) | 0.683 |
| Body mass index, kg/m$^2$ | 27.99 | 28.01 | . . . | 27.84 (24.34; 31.70) | 28.33 (23.16; 31.47) | 0.445 | 28.96 ± 4.59 | 27.86 ± 3.58 | 0.281* |
| Height, cm | 165.40 (160.20; 72.60) | 166.30 (158.20;175.10) | 0.691 | 161.07 ± 5.49 | 159.29 ± 7.25 | 0.244* | 174.25 ± 6.22 | 176.20 ± 7.46 | 0.178* |
| Weight, kg | 78.70 (67.40; 89.90) | 78.60 (66.30; 90.50) | 0.628 | 72.70 (64.05; 80.95) | 69.80 (59.30; 79.50) | 0.222 | 87.70 (79.27; 97.20) | 86.90 (78.70; 92.68) | 0.674 |
| Serum 25(OH)D, ng/ml | 23.84 (17.31; 31.40) | 24.90 (17.21; 33.35) | 0.733 | 25.30 (19.41; 33.40) | 32.87 (24.68; 37.85) | 0.065 | 21.24 (16.48; 26.75) | 18.59 (16.55; 25.12) | 0.348 |
| Whole-body BMD, g/cm$^2$ | 1.10 ± 0.14 | 1.08 ± 0.13 | 0.266* | 1.03 ± 0.12 | 1.00 ± 0.12 | 0.246* | 1.20 ± 0.11 | 1.16 ± 0.10 | 0.113* |
| Whole-body T-score | −0.28 ± 1.18 | −0.60 ± 1.10 | 0.079* | −0.47 ± 1.20 | −0.77 ± 1.18 | 0.246* | −0.02 ± 1.09 | −0.42 ± 0.99 | 0.104* |
| Lumbar spine BMD, g/cm$^2$ | 1.14 (0.99; 1.28) | 1.12 (1.01; 1.26) | 0.716 | 1.03 (0.94; 1.17) | 1.07 (0.92; 1.14) | 0.970 | 1.26 ± 0.17 | 1.21 ± 0.17 | 0.212* |
| Femoral neck BMD, g/cm$^2$ | 0.91 (0.82; 0.99) | 0.88 (0.79; 0.91) | **0.031** | 0.87 (0.78; 0.96) | 0.81 (0.77; 0.89) | 0.058 | 0.96 (0.88; 1.03) | 0.90 (0.84; 0.98) | 0.084 |
| Femoral neck T-score | −1.08 (−1.66; −0.46) | −1.45 (−1.92; −0.94) | **0.015** | −1.20 (−1.84; −0.53) | −1.61 (−1.94; −1.04) | 0.058 | −0.88 (−1.46; −0.30) | −1.28 (−1.73; −0.73) | 0.082 |
| Total femoral BMD, g/cm$^2$ | 1.00 (0.90; 1.10) | 0.95 (0.89; 1.02) | 0.060 | 0.95 ± 0.14 | 0.92 ± 0.13 | 0.325 | 1.07 (0.99; 1.16) | 0.96 (0.92; 1.09) | **0.012*** |
| Total femur T-score | −0.37 (−1.09; 0.33) | −0.78 (−1.27; −0.24) | **0.028** | −0.44 ± 1.11 | −0.67 ± 1.01 | 0.325 | −0.23 (−0.75; 0.42) | −0.96 (−1.27; −0.09) | **0.011*** |

* Variables with normal distributions were compared using the Student t-test, and those with non-normal distributions were compared using Wilcoxon's test for independent samples.

AMD-0: group without AMD; AMD-1: group with AMD.

AMD, age-related macular degeneration; BMD, bone mineral density.

                                                  

**Table 2. The relationship between T-Score (Cutoff, −1.0) for femoral BMD and the occurrence of age-related macular degeneration in women and men.**

| Variable | Women | | | | Men | | | |
|---|---|---|---|---|---|---|---|---|
| | AMD-0 (n = 223) | AMD-1 (n = 23) | Odds ratio (95% CI) | p | AMD-0 (n = 158) | AMD-1 (n = 22) | Odds ratio (95% CI) | p |
| Total femur T-score" ≥ −1.0 | 144 (64.6) | 16 (69.6) | 0.80 (0.31; 2.02) | 0.633 | 134 (84.8) | 12 (54.5) | 4.65 (1.81; 11.97) | **0.001** |
| Total femur T-score" < -1.0 | 79 (35.4) | 7 (30.4) | | | 24 (15.2) | 10 (45.5) | | |

AMD-0: group without AMD; AMD-1: group with AMD.

AMD, age-related macular degeneration; CI, confidence interval.

The proportions of men in the AMD-0 and AMD-1 groups who had decreased T-scores for femoral BMD were 15% and 46%, respectively, demonstrating a statistically significant difference (OR = 4.65 (1.81; 11.97); p = 0.001) The presence of AMD did not exert a significant effect on the occurrence of decreased T-score values among women (p = 0.633), (Table 2).

Femoral BMD was a significant predictor of AMD in men based on univariate logistic regression models. A one-unit increase in femoral BMD was associated with a 98% decrease in the risk of being in the AMD-1 group (95% CI = <0.01; 0,80; p = 0.043), whereas a one-unit increase in the femur T-score was associated with a 42% risk reduction (95% CI = 0.34; 0.96; p = 0.041)). For women only age was a significant predictor of AMD, the odds for AMD were growing with age, one-year growth was increasing odds for AMD by 8% (95% CI = 1.02; 1.15; p = 0.012). Femoral BMD as well as femur T-score were not significantly associated with occurrence of AMD in women (p = 0.352 in both cases), (Table 3).

Two multivariable logistic regression models were created for women and men, both with AMD as the outcome variable: the predictors were age and crude femoral BMD in one (Table 4), and age and femur T-score in the other (Table 5). For men femoral BMD and femur T-score were both significant predictors, whereas age was not significant in either model (p = 0.409 for the model with BMD and p = 0.406 for the model with the T-score). The Nagelkerke pseudo-$R^2$ for both multivariable models and the univariate models, including crude values of BMDs and T-scores, was 0.05, indicating that the inclusion of age in the multivariable models did not improve their goodness of fit. However, the Nagelkerke pseudo-$R^2$ was low, implying that the occurrence of AMD in men was influenced by variables not included in the models. Based on the chi-square statistics, none of the multivariable models that included age were statistically significant (p = 0.074 for the model with T-score and p = 0.080 for the model with BMD). For women none of variables was significant in both models (p > 0.05 in all cases) indicating that both multivariate models for women were not significant.

## Discussion

Our study revealed an association between a decrease in femoral BMD and the occurrence of AMD in men older than 50 years. In contrast, we did not observe any differences in the

**Table 3. Univariate logistic regression models with age-related macular degeneration as an outcome variable in women and men.**

| Variable | Women | | | Men | | |
|---|---|---|---|---|---|---|
| | Odds ratio | 95% Confidence interval | p | Odds ratio | 95% Confidence interval | p |
| **Total femoral BMD** | 0.22 | 0.01; 5.03 | 0.352 | 0.02 | <0.01; 0.80 | **0.043** |
| **Total femur T-score** | 0.82 | 0.54; 1.23 | 0.352 | 0.58 | 0.34; 0.96 | **0.041** |
| **Age** | 1.08 | 1.02; 1.15 | **0.012** | 0.98 | 0.92; 1.04 | 0.502 |

BMD, bone mineral density.

**Table 4. Multivariable logistic regression model with age-related macular degeneration as an outcome variable and age and femoral BMD as independent variables in women and men.**

| Variable | Women | | | Men | | |
|---|---|---|---|---|---|---|
| | Odds ratio | 95% Confidence interval | p | Odds ratio | 95% Confidence interval | p |
| Total femoral BMD | 0.40 | 0.01; 9.35 | 0.576 | 0.02 | <0.01; 0.70 | **0.037** |
| Age | 1.06 | 0.98; 1.13 | 0.0504 | 0.98 | 0.92; 1.03 | 0.409 |

BMD, bone mineral density.

densitometric parameters of women in relation to AMD prevalence. The results of our study provide grounds for actively screening for AMD among patients with osteopenia. Routine screening diagnostics in the form of the Amsler grid eye test should be considered in patients with low bone density.

Published data on the coincidence of osteoporosis and AMD are scarce. The first such study included only Caucasian women aged >75 years from the USA [5], in which women with a high hip BMD presented with concomitant AMD less often than those with a lower BMD. In that study, the average patient age was 80 years; therefore, the results may not be generalizable to younger women. The authors suggested an inverse association between BMD and AMD, which may be caused by the protective effect of estrogen against AMD. In the second study, conducted in a Korean population [6], a linear relationship was discovered between the osteoporosis status of the femoral neck and AMD in female participants, but showed no significant relationship between the co-occurrence of osteoporosis with either early or late AMD in men. The authors of that report suggested that the lack of an association among men might have been due to the low prevalence of osteoporosis and AMD in the male population. They also suggested that calcium and phosphate deposited in the bloodstream during bone transformation in osteoporosis may initiate drusen formation and contribute to the development of early AMD. Moreover, they discovered a strong correlation between early AMD and osteoporosis, whereas this correlation did not occur in late stages of AMD [16]. In our study only men had significant association between AMD and osteoporosis. This may result from the fact that, in the Korean study, the cutoff point used for the T-score was −2.5, whereas, in our study, we included patients with osteoporosis and osteopenia in the low bone density group (T-score <−1.0). Although they didn't reveal a relation between osteoporosis and AMD in men, they demonstrated a significantly lower hip and femoral neck BMD for men with AMD than for men without AMD. In the study from the USA the median age of women was several years older than that in our study, which may explain the differences in results.

The pathogenesis of AMD is multifactorial, and the molecular mechanisms underlying its development are still not understood. Its pathogenesis involves environmental and genetic factors, disorders of the complement system, lipid abnormalities, angiogenic and inflammatory pathways, and the extracellular matrix [16, 17]. A growing body of evidence suggests that oxidative stress and impaired protein degradation and clearance pathways in retinal pigment

**Table 5. Multivariable logistic regression model with age-related macular degeneration as an outcome variable and age and femur T-score as independent variables in women and men.**

| Variable | Women | | | Men | | |
|---|---|---|---|---|---|---|
| | Odds ratio | 95% Confidence interval | p | Odds ratio | 95% Confidence interval | p |
| Femur T-score | 0.89 | 0.58; 1.33 | 0.576 | 0.57 | 0.33; 0.94 | **0.035** |
| Age | 1.06 | 0.95; 1.13 | 0.051 | 0.98 | 0.92; 1.03 | 0.406 |

epithelial (RPE) cells may lead to their functional impairment and damage. Autophagy plays a key role in eliminating cellular waste materials, such as aggregated proteins and dysfunctional cell organelles, from RPE cells. RPE cells cultured from donors with AMD are reportedly more sensitive to oxidative stress and impairment of autophagy than RPE cells from donors without AMD [18]. Autophagy is vital for the survival of cells under stress, especially RPE cells, given their metabolic activity and paucity of physiological functions. Impairment of autophagy leads to lipofuscin accumulation in RPE cells and deposition of lipoproteins, in the form of drusen, in the extracellular spaces [18, 19].

One possible reason for the coexistence of AMD with bone loss or decreased BMD is the impairment of autophagy observed in both conditions. Both AMD and senile osteoporosis are associated with decreased autophagic activity. Impaired autophagy plays a pivotal role in the onset and progression of osteoporosis [18]. Although the underlying mechanisms remain unclear, another common feature of osteoporosis and AMD is the p62 protein encoded by the *SQSTM1* gene (p62/SQSTM1). Intracellular p62 is associated with autophagy, bone remodeling, bone marrow integrity, cancer progression, and the maintenance of systemic homeostasis. P62/SQSTM1 acts as an effective suppressor of inflammation by reducing oxidative stress and proinflammatory signals. It is involved in the elimination of proteins from RPE cells via autophagy. Owing to impairment of autophagy, an accumulation of p62/SQSTM1-labeled waste material has been observed in patients with AMD, which leads to the activation of inflammasome signaling in conjunction with IL-1β secretion [20]. P62/SQSTM1 also contributes to an increase in the secretion of IL-8, the most potent chemotactic factor, via IL-1β stimulation in RPE cells. Patients with AMD have elevated p62/SQSTM1 levels in the retina. While the foveomacular area in such patients reportedly exhibit higher p62/SQSTM1 staining than the perimacular and peripheral areas, no regional differences were observed in persons without AMD [21]. Oxidative stress reportedly results in the increase of p62/SQSTM1 mRNA and protein levels, which activates autophagy and constitutes a protective mechanism against oxidative stress-induced cell death. In turn, the downregulation of p62/SQSTM1 reportedly decreases the sensitivity of RPE cells to oxidative stress, as it reduces autophagic degradation [22].

P62/SQSTM1 may also be involved in bone diseases, as a high frequency of *SQSTM1* gene mutations were reportedly observed in patients with Paget's disease [23, 24]. Impaired activity of the p62/SQSTM1 protein upregulates osteoclast functioning [25]. Moreover, a DNA plasmid encoding p62/SQSTM1 (p62 DNA) exhibited strong anti-osteoporotic activity in animal models [26]. Furthermore, the retinoprotective effect of p62 DNA in AMD warrants mention [27]. This plasmid decreases the severity of retinopathy, slows the progression of destructive alterations within RPE cells, and inhibits microglia/macrophage migration to the outer retina in senescence-accelerated OXYS rat models. Additionally, sharp downregulation of master proinflammatory cytokines and upregulation of endogenous p62 protein have been observed after administration of p62 DNA [26].

The prevalence of AMD in our study was 10.8%. Although the global prevalence of AMD is estimated at 8.7%, a meta-analysis has suggested that its prevalence is higher in European countries (12.3%–18.3%) [28].

Our study is not without limitations. First, it was limited by its observational nature, relatively small number of patients, and lack of detailed data on AMD status. Second, we did not analyze the effects of other factors involved in the coincident development of AMD and osteoporosis, e.g., smoking, alcohol consumption, and dietary calcium intake. We did not have access to fragility fracture data or a fall registry for the study population. Furthermore, the health data of the study participants were obtained at a single timepoint. As certain parameters, such as BMD, body mass index, calcium intake, and serum vitamin D level, change over time, this factor should be considered a potential confounder. Third, the group of women with

AMD was older than that without AMD. As the risks of AMD and osteoporosis both increase with age, this factor may explain the lack of a significant relationship between AMD and low BMD in women in this study. Differences in BMD between men with and those without AMD were detected only in the femoral region and not in the lumbar spine. We could not further explore this association, as dual-energy X-ray absorptiometry does not provide insights into bone quality and structure. Despite these limitations, this is the first study in which the link between a low hip BMD and AMD was demonstrated in a Polish population.

In conclusion, further longitudinal research with a larger group of participants is required to confirm the link and potential pathogenic association between AMD and a low BMD in the Polish population.

## Author Contributions

**Conceptualization:** Karol Kamiński, Joanna Konopińska.

**Data curation:** Marcin Palewski, Magda Łapińska.

**Formal analysis:** Agnieszka Budnik, Magdalena Michnowska-Kobylińska, Łukasz Lisowski, Zofia Stachurska, Karol Kamiński.

**Methodology:** Anna Szpakowicz, Jerzy Konstantynowicz, Karol Kamiński.

**Supervision:** Anna Szpakowicz.

**Writing – original draft:** Agnieszka Budnik.

**Writing – review & editing:** Anna Szpakowicz, Jerzy Konstantynowicz, Karol Kamiński, Joanna Konopińska.

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
