## [Decision Letter · Decision Letter 0]

24 Jul 2023

PONE-D-23-18141

The prevalence of age-related macular degeneration and osteoporosis in the older Polish population: is there a link?

PLOS ONE

Dear Dr. Konopińska,

Thank you for submitting your manuscript to PLOS ONE. After careful consideration, we feel that it has merit but does not fully meet PLOS ONE’s publication criteria as it currently stands. Therefore, we invite you to submit a revised version of the manuscript that addresses the points raised during the review process.

We look forward to receiving your revised manuscript.

Kind regards,

Tatsuya Inoue

Academic Editor

PLOS ONE

Journal Requirements:

Reviewers' comments:

Reviewer's Responses to Questions

**Comments to the Author**

1. Is the manuscript technically sound, and do the data support the conclusions?

Reviewer #1: Partly

Reviewer #2: Yes

2. Has the statistical analysis been performed appropriately and rigorously? 

Reviewer #1: I Don't Know

Reviewer #2: Yes

3. Have the authors made all data underlying the findings in their manuscript fully available?

Reviewer #1: Yes

Reviewer #2: No

4. Is the manuscript presented in an intelligible fashion and written in standard English?

Reviewer #1: Yes

Reviewer #2: Yes

5. Review Comments to the Author

Reviewer #1: Please see the attachment-------------------------------------------------------------------------------------------------------------------------------------------------------------------------------------------------------------------

Reviewer #2: This is generally a well planned work. However, there are a few of observations which when addressed will greatly improved the quality of this research presentation.

Line 36-56 - kindly breakdown your abstract into ; Background, Objectives, Methods, Results and conclusion.

Line 71-72 - Did the cohort study from Australia find the predisposing factors for in ARM? Again, what if the poor vision which commonly predisposes to fall which can cause fractures.

Line 80-82 - please apart from Geography and ethnicity, is there any contributory factor to osteoporosis in the countries mentioned?

Line 84-85 - Please did your literature source state why prevalence of osteoporosis is underestimated in men?

Line 87-88 - why not say 'Estimates from developed countries suggest ...'

Line 91-93 - This point is the reason for the comment on line 71-72

Tables 4,5 & 6 has various variables of interest among men alone.

I suggest analysis for same variable among women should be done and reported. For each variable, results should be merged and clearly labeled 'men and women'

Table 2 - 6 it appears authors reported analysis of different variables for men and women.

Suggestion, authors should analyse the same variable for men and women and show results on same table.

Example

Table 2 shows relationship between T-score (cut off, --1.0) for femoral BMD and occurrence of AMD in men.

It is actually possible to report same in women and present in the same table.

The table can be labeled relationship between T-Score (cut off, -1.0) for femoral BMD and occurrence of AMD in men and women.

I suspect table 2 & 3 are the same for men and women. If they are the same, it's important to note that the labels are different. This ought not be so.

If they are the same, they should be merged.

There should be results with the same headings for tables 4, 5 & 6 among males and females. Also the tables should be merged with clear labels for men and women.

Suggested Analysis - Authors should consider running a correlation analysis for BMD/osteoporosis and occurrence of ARMD in both sexes.

Like I said before, if these are implemented the quality of your presentation will further improve.

6. PLOS authors have the option to publish the peer review history of their article (what does this mean?). If published, this will include your full peer review and any attached files.

Reviewer #1: No

Reviewer #2: No

---

## [Author Response · Author response to Decision Letter 0]

5 Sep 2023

23/AUG/2023

Emily Chenette

Editor-in-Chief

PLoS One

Dear Editor: 

I wish to re-submit the manuscript titled “The prevalence of age-related macular degeneration and osteoporosis in the older Polish population: is there a link?” The manuscript ID is PONE-D-23-18141.

We thank you and the reviewers for your thoughtful suggestions and insights. The manuscript has benefited from these insightful suggestions. I look forward to working with you and the reviewers to move this manuscript closer to publication in PLoS One.

The manuscript has been rechecked and the necessary changes have been made in accordance with the reviewers’ suggestions. The responses to all comments have been prepared and attached herewith. All of the relevant changes in the revised manuscript have been indicated with red font.

We would like to add information regarding to the founding as follows: “This study was supported by funds of the Ministry of Science and Higher education (Poland) within the project "The Excellence Initiative - Research University" (proteomic research), and from statutory funds of Medical University of Bialystok.”

We have also publicated the data for our study in repositorium on the following link:

https://dataverse.harvard.edu/dataset.xhtml?persistentId=doi:10.7910/DVN/TZVNHA

Thank you for your consideration. I look forward to hearing from you.

Sincerely,

Joanna Konopińska

Medical University of Białystok

Kilinskiego 1 STR 15-201

Poland

joannakonopinska@o2.pl

Phone:

+48857468372

E-mail: joannakonopinska@o2.pl

Responses to the Reviewers’ comments

Dear Reviewers:

We would like to thank you for the detailed review of our manuscript and your valuable remarks. The manuscript has been rechecked and the necessary changes have been made in accordance with your suggestions. All changes were highlighted in red font. The responses to all comments are given below. We hope that you will find our explanations and manuscript modifications satisfactory for publication in PLoS One.

Reviewer 1

Comments to the Authors.

This handles an interesting theme regarding the association between the osteoporosis and AMD. There are several comments and questions.

Response:

Thank you for this comment, we appreciate it!

Major points:

1. Comment: How relevant the findings are for practitioners and patients? That is, what impact will the findings have on the current clinical practice? The authors should discuss about it.

Response: Thank you for pointing out this issue, you are right as we did not sufficiently emphasize the importance of the study results for practisioners. We have added a relevant paragraph about it in the Discussion, as follows: “The results of our study provide grounds for actively screening for AMD among patients with osteopenia. Routine screening diagnostics in the form of the Amsler eye grid test should be considered in patients with low bone density.” (lines 250-253)

2. Comment: Line 337: “As the risks of AMD and osteoporosis both increase with age, this factor may explain the lack of a significant relationship between AMD and low BMD in women in this study.”

This explains why there was no significant association between AMD and osteoporosis for women. What do the authors think about only men having significant association? The findings seem to be opposite to the previous reports from the USA and Korea the authors cited.

Response: : We agree. This is an important point. We have added the clarification as follows: “This may result from the fact that, in the Korean study, the cutoff point used for the T-score was −2.5, whereas, in our study, we included patients with osteoporosis and osteopenia in the low bone density group (T-score <−1.0). Although they didn't reveal a relation between osteoporosis and AMD in men, they demonstrated a significantly lower hip and femoral neck BMD for men with AMD than for men without AMD. In the study from the USA the median age of women was several years older than that in our study, which may explain the differences in results”. (lines 249-256)

3. Comment: Were there any data on cataract, glaucoma and diabetic retinopathy? The association between these disorders and osteoporosis would be valuable, which would clarify if the current findings are a disease-specific association, that is, if only AMD is associated with osteoporosis. 

Response:

Thank you for your comment. Unfortunately, we did not collect any data regarding cataracts, glaucoma, or diabetic retinopathy in this study. Little data to this effect have been reported. Although cataracts and AMD are different conditions, they can coexist in the same individual owing to the shared risk factor of age. Interestingly, glaucoma and AMD share some risk factors. Both glaucoma and AMD are more common in older individuals. Moreover, family history and genetics play a role in the development of these diseases. Some studies suggest that cardiovascular factors, such as high blood pressure and atherosclerosis, may contribute to the development of these pathologies. In addition, smoking is a significant risk factor for AMD and has also been associated with an increased risk of glaucoma. Having diabetes can increase the risk of developing other eye conditions and complications, including cataracts, glaucoma, and possibly AMD. However, the development of AMD is not solely due to diabetic retinopathy. Despite these shared risk factors, the underlying mechanisms and pathophysiology of glaucoma, cataracts, and AMD have not yet been clearly identified.

Huang HK, Lin SM, Loh CH, Wang JH, Liang CC. Association Between Cataract and Risks of Osteoporosis and Fracture: A Nationwide Cohort Study. J Am Geriatr Soc. 2019 Feb;67(2):254-260. doi: 10.1111/jgs.15626. Epub 2018 Oct 3. PMID: 30281143.

Roberts SB, Silver RE, Das SK, Fielding RA, Gilhooly CH, Jacques PF, Kelly JM, Mason JB, McKeown NM, Reardon MA, Rowan S, Saltzman E, Shukitt-Hale B, Smith CE, Taylor AA, Wu D, Zhang FF, Panetta K, Booth S. Healthy Aging-Nutrition Matters: Start Early and Screen Often. Adv Nutr. 2021 Jul 30;12(4):1438-1448. doi: 10.1093/advances/nmab032. Erratum in: Adv Nutr. 2021 Jul 30;12(4):1597-1598. PMID: 33838032; PMCID: PMC8994693.

Minor point:

1. Comment: Introduction is too long. The authors might want to reduce it below 300 words. General explanation of the disease would be unnecessary like “Clinically, osteoporosis manifests as a higher risk of skeletal fragility and apparent major osteoporotic fractures that frequently lead to disability, mobility restriction, decreased quality of life, and premature mortality.”

Response:

Thank you for this comment. The introduction has been shortened, accordingly.

2. Comment: Line 223-233: “the predictors were age and crude femoral BMD in one (Table 5), and age and femur T-score in the other (Table 6).” This should be removed to the Method section. 

Response:

Thank you for this suggestion. The paragraph has been moved to the Methods section.

3. Comment: The authors should describe why the threshold (T-score = -0.1) was selected (e.g. it’s clinically important, often used, etc.). 

Response:

We are terribly sorry for this typographical error. The T-score has been set to −1.0. The mistake has been corrected throughout the manuscript.

4. Comment: What is the difference between the “Total femur T-score” in the bottom row of Table 1 and “T-score≥-0.1” or “T-score<-0.1” in Table 2? Is something just categorized “Total femur T-score” into used in Table 2? Why did the authors take the trouble to do so? This is related to the comment 3.

Response:

We apologize for being imprecise. “T-score” in Table 2 has been corrected to “Total femur T-score.”

5. Comment: I couldn’t understand the part of Nagelkerke pseudo-R2. The authors should explain how to interpret this statistic value for readers to be able to understand easily in the Methods section (not in the Results section).

Response:

For clarification purposes we have added the explanation as follows’ and we have moved this part to the Method section

6. Comment: I think the following paragraph is redundant and not necessary (lines 265-272): “The first manifestation of AMD is damage to the retinal pigment epithelium (RPE), a monolayer of cuboidal, pigmented epithelial cells located between the neurosensory retina and the choroid. It has a plethora of functions, including delivering blood-derived nutrients to photoreceptors; transporting ions, water, and metabolites from the subretinal space to the blood; absorbing light; phagocytizing photoreceptor outer segments; supporting recycling of all‐ trans retinal back to 11‐ cis retinal; secreting neurotrophic factors; and scavenging damaged reactive oxygen species [19]. Thus, functional abnormalities in the RPE lead to impaired function of the entire neural retina.” 

Response:

Thank you for this significant comment. We agree and removed that paragraph from the revised manuscript.

7. Comment: I think the following paragraph is so long and should be made short (lines 273-294): “The pathogenesis of AMD is multifactorial, and the molecular mechanisms underlying its development are still not understood. Its pathogenesis involves environmental and genetic factors, disorders of the complement system, lipid abnormalities, angiogenic and inflammatory pathways, and the extracellular matrix [19,20]. A growing body of evidence suggests that oxidative stress and impaired protein degradation and clearance pathways in RPE cells may lead to their functional impairment and damage. Autophagy plays a key role in eliminating cellular waste materials, such as aggregated proteins and dysfunctional cell organelles, from RPE cells. RPE cells cultured from donors with AMD are reportedly more sensitive to oxidative stress and impairment of autophagy than RPE cells from donors without AMD [21]. Autophagy is vital for the survival of cells under stress, especially RPE cells, given their metabolic activity and paucity of physiological functions. Impairment of autophagy leads to lipofuscin accumulation in RPE cells and deposition of lipoproteins, in the form of drusen, in the extracellular spaces. Drusen formation is the first manifestation of one of the two forms of AMD, namely, dry AMD (dAMD). dAMD is also characterized by RPE dysfunction and photoreceptor loss. In approximately 10%–15% of cases, dAMD progresses to wet AMD (wAMD), characterized by choroidal neovascularization with intra- and subretinal shunting, hemorrhages, and RPE detachment. wAMD can be treated with intravitreal injections of antiVEGF agents. Although this form of AMD accounts for only 10%–15% of all AMD cases, it is the leading cause of substantial eyesight deterioration and the principal source of expenditures associated with treatment for AMD [21,22]. A conservative estimate of the cost of wAMD treatment in the USA was demonstrated to generate a cost of $1.1 billion for the entire population in year 1 and $5.1 billion in year 3 [22].”

The next paragraph refers to “autophagy”, which is only the element needed in this paragraph (lines 273-294).

Response: Thank you for the suggestion. We agree and shortened that paragraph, accordingly.

8. Comment: Line 342: “Nonetheless, even considering that the vertebral BMD in these individuals remained unaffected, a lower femoral BMD may increase the risk of osteoporosis and subsequent hip fractures in men”

I couldn’t understand this sentence. I thought the authors investigated the association between AMD and osteoporosis (T-score), not hip fractures. Did the authors look into hip fractures?

Response:

We apologize for the confusion. We indeed did not investigate hip fractures. We have removed this sentence.

Reviewer 2

1. Comment: This is generally a well-planned work. However, there are a few of observations which when addressed will greatly improve the quality of this research presentation.

Response:

Thank you very much for this comment. We appreciate it.

2. Comment: Line 36-56 - kindly breakdown your abstract into ; Background, Objectives, Methods, Results and conclusion.

Response:

Thank you for this comment. The abstract has been structured accordingly

3. Comment: Line 71-72 - Did the cohort study from Australia find the predisposing factors for in ARM? Again, what if the poor vision which commonly predisposes to fall which can cause fractures.

Response: Taking into account the higher risk of falls in patients with poor vision, we want to emphasize that those falls in the AMD group resulted in fractures characteristic of osteoporosis. The Australian study showed that, apart from ARM, cataracts and corneal diseases were associated with a high risk of spinal and humero-radio-ulnar fractures. 

4. Comment: Line 80-82 - please apart from Geography and ethnicity, is there any contributory factor to osteoporosis in the countries mentioned?

Response:

Osteoporosis is a multifactorial disease, and its prevalence and risk factors can vary significantly between different countries and populations. While some factors may be universal, others may be more prevalent in specific regions owing to cultural, environmental, or genetic differences. First, ethnicity and genetic variations can influence bone mineral density and fracture risk, leading to differences in osteoporosis prevalence across populations. Moreover, dietary patterns can differ significantly between countries, and nutrition plays a crucial role in bone health. Countries with diets rich in calcium and vitamin D tend to have better bone health. In addition, levels of physical activity and exercise habits can vary by culture and lifestyle. Regular weight-bearing exercises are proven to be beneficial for bone health, and countries with more active populations may have lower osteoporosis rates. Interestingly, certain hormonal factors, such as early menopause or hormonal disorders, can increase the osteoporosis risk. These factors can be influenced by genetics and lifestyle and may differ between populations. Next, countries with higher rates of alcohol and tobacco consumption may have an increased prevalence of osteoporosis, as both these substances can negatively impact bone health. Moreover, populations with different obesity rates may experience varying impacts on bone health. Both very-low and very-high BMI levels can be associated with an increased risk of osteoporosis. Access to healthcare and awareness about osteoporosis and its prevention can affect its prevalence. Countries with better healthcare systems and osteoporosis education programs may have lower rates of the disease. Importantly, these factors can interact and have complex relationships with each other. The prevalence of osteoporosis in different countries can also be influenced by the aging population and changes in lifestyle and dietary habits over time.

Chotiyarnwong P, McCloskey EV, Harvey NC, Lorentzon M, Prieto-Alhambra D, Abrahamsen B, Adachi JD, Borgström F, Bruyere O, Carey JJ, Clark P, Cooper C, Curtis EM, Dennison E, Diaz-Curiel M, Dimai HP, Grigorie D, Hiligsmann M, Khashayar P, Lewiecki EM, Lips P, Lorenc RS, Ortolani S, Papaioannou A, Silverman S, Sosa M, Szulc P, Ward KA, Yoshimura N, Kanis JA. Is it time to consider population screening for fracture risk in postmenopausal women? A position paper from the International Osteoporosis Foundation Epidemiology/Quality of Life Working Group. Arch Osteoporos. 2022 Jun 28;17(1):87. doi: 10.1007/s11657-022-01117-6. PMID: 35763133; PMCID: PMC9239944.

5. Comment: Line 84-85 - Please did your literature source state why prevalence of osteoporosis is underestimated in men?

Response:

Yes, the literature states that: “Osteoporosis is commonly recognized as a disease in women following menopause, which is often overlooked in men mainly because there is no aging process in men analogous to menopause with a resultant rapid loss of bone mass. In men, secondary osteoporosis is more frequent; common causes include glucocorticoid excess, hypogonadism, and alcohol abuse.”A relevant high quality references have been now included in the revised version.

Vilaca T, Eastell R, Schini M. Osteoporosis in men. Lancet Diabetes Endocrinol. 2022 Apr;10(4):273-283. doi: 10.1016/S2213-8587(22)00012-2. Epub 2022 Mar 2. PMID: 35247315.

6. Comment: Line 87-88 - why not say 'Estimates from developed countries suggest ...'

Response:

The introduction was shortened to within 300 words, according to a suggestion of Reviewer 1. The related sentence now states: “Every third woman and every fifth man over the age of 50 years in developed countries are estimated to experience a bone fracture during their remaining lifetime” We hope that this is suitable. 

7. Comment:

Line 91-93 - This point is the reason for the comment on line 71-72

Tables 4,5 & 6 has various variables of interest among men alone.

I suggest analysis for same variable among women should be done and reported. For each variable, results should be merged and clearly labeled 'men and women'

Response:

Data for women were added to tables 4,5,6 and labels were corrected accordingly.

8. Comment: Table 2 - 6 it appears authors reported analysis of different variables for men and women. Suggestion, authors should analyse the same variable for men and women and show results on same table.

Example

Table 2 shows relationship between T-score (cut off, --1.0) for femoral BMD and occurrence of AMD in men.

It is actually possible to report same in women and present in the same table.

The table can be labeled relationship between T-Score (cut off, -1.0) for femoral BMD and occurrence of AMD in men and women.

9. Comment: I suspect table 2 & 3 are the same for men and women. If they are the same, it's important to note that the labels are different. This ought not be so.

If they are the same, they should be merged.

Response:

This is correct. Tables 2 and 3 are the same with data for men in table 2 and data for women in table 3. Both tables were merged into one table (2) and original table 3 was removed.

10. Comment: There should be results with the same headings for tables 4, 5 & 6 among males and females. Also, the tables should be merged with clear labels for men and women.

Response:

Re. your comments and suggestions specified above. We agree, and we have made appropriate revisions accordingly. The data have been presented for both women and men when possible. And therefore, some data in the tables have been merged and modified. 

11. Comment: Suggested Analysis - Authors should consider running a correlation analysis for BMD/osteoporosis and occurrence of ARMD in both sexes.

Response: Thank you for this suggestion. We are aware that there is, in fact, a strong connection between BMD (crude values, Z-scores and T-scores) and osteoporosis, given that the diagnoses of ‘osteopenia’ and ‘osteoporosis’ are based just on BMD T-score values. This analysis is included in tables 2 ad 3 of initial manuscript (table 2 of current version).

Like I said before, if these are implemented the quality of your presentation will further improve.

---

## [Decision Letter · Decision Letter 1]

6 Oct 2023

The prevalence of age-related macular degeneration and osteoporosis in the older Polish population: is there a link?

PONE-D-23-18141R1

Dear Dr. Konopińska,

We’re pleased to inform you that your manuscript has been judged scientifically suitable for publication and will be formally accepted for publication once it meets all outstanding technical requirements.

Kind regards,

Tatsuya Inoue

Academic Editor

PLOS ONE

Additional Editor Comments (optional):

Reviewers' comments:

Reviewer's Responses to Questions

**Comments to the Author**

1. If the authors have adequately addressed your comments raised in a previous round of review and you feel that this manuscript is now acceptable for publication, you may indicate that here to bypass the “Comments to the Author” section, enter your conflict of interest statement in the “Confidential to Editor” section, and submit your "Accept" recommendation.

Reviewer #2: All comments have been addressed

2. Is the manuscript technically sound, and do the data support the conclusions?

Reviewer #2: Yes

3. Has the statistical analysis been performed appropriately and rigorously? 

Reviewer #2: Yes

4. Have the authors made all data underlying the findings in their manuscript fully available?

Reviewer #2: Yes

5. Is the manuscript presented in an intelligible fashion and written in standard English?

Reviewer #2: Yes

6. Review Comments to the Author

Reviewer #2: Happy with the responses

7. PLOS authors have the option to publish the peer review history of their article (what does this mean?). If published, this will include your full peer review and any attached files.

Reviewer #2: No

---

## [Editor Report · Acceptance letter]

11 Oct 2023

PONE-D-23-18141R1 

The prevalence of age-related macular degeneration and osteoporosis in the older Polish population: is there a link? 

Dear Dr. Konopińska:

I'm pleased to inform you that your manuscript has been deemed suitable for publication in PLOS ONE. Congratulations! Your manuscript is now with our production department. 

Kind regards, 

on behalf of

Dr. Tatsuya Inoue 

Academic Editor

PLOS ONE